**Data Availability Statement:** All relevant data are within the manuscript.

# Clinical course and short-term outcome of postsplenectomy reactive thrombocytosis in children without myeloproliferative disorders: A single institutional experience from a developing country

**Zlatan Zvizdic**[1], **Aladin Kovacevic**[2], **Emir Milisic**[1], **Asmir Jonuzi**[1], **Semir Vranic**[3]*

**1** Clinic of Pediatric Surgery, University Clinical Center Sarajevo, Sarajevo, Bosnia and Herzegovina, **2** Public Health Institution "Community Health Center Jajce", Jajce, Bosnia and Herzegovina, **3** College of Medicine, QU Health, Qatar University, Doha, Qatar

* semir.vranic@gmail.com, svranic@qu.edu.qa

## Abstract

### Objectives

To evaluate the clinical outcome and complications in the pediatric population who had splenectomy at our institution, emphasizing the incidence of postplenectomy reactive thrombocytosis (RT) and its clinical significance in children without underlying hematological malignancies.

### Materials and methods

The medical records of pediatric patients undergoing splenectomy were retrospectively reviewed for the period 1999–2018. The following variables were analyzed: Demographic parameters (age, sex), indications for surgery, operative procedures, preoperative and postoperative platelet count (postplenectomy RT), the use of anticoagulant therapy, and postoperative complications. The patients were divided into two groups according to indications for splenectomy: The non-neoplastic hematology group and the non-hematology group (splenectomy for trauma or other spleen non-hematological pathology).

### Results

Fifty-two pediatric (37 male and 15 female) patients who underwent splenectomy at our institution were reviewed. Thirty-four patients (65%) were in the non-hematological group (splenic rupture, cysts, and abscess) and 18 patients (35%) in the non-neoplastic hematological group (hereditary spherocytosis and immune thrombocytopenia). The two groups did not differ significantly in regards to the patients' age, sex, and preoperative platelet count ($P$>0.05 for all variables). Forty-nine patients (94.2%) developed postplenectomy RT. The percentages of mild, moderate and extreme thrombocytosis were 48.9%, 30.7%, and 20.4%, respectively. The comparisons of RT patients between the non-neoplastic

**Funding:** The publication of this article was funded by the Qatar National Library. The funder had no role in study design, data collection and analysis, decision to publish, or preparation of the manuscript.

**Competing interests:** The authors have declared that no competing interests exist.

**Abbreviations:** AML, Acute myelogenous leukemia; CML, Chronic myeloid leukemia; HS, Hereditary spherocytosis; IL, Interleukin(s); INR, International normalized ratio; ITP, Immune thrombocytopenia; PSVT, Portal splenic vein thrombosis; RT, Reactive thrombocytosis; SW test, Shapiro-Wilk test.

hematology and the non-hematology group revealed no significant differences in regards to the patients' age, sex, preoperative and postoperative platelet counts, preoperative and postoperative leukocyte counts, and the average length of hospital stay ($P>0.05$ for all variables). None of the patients from the cohort was affected by any thrombotic or hemorrhagic complications.

## Conclusions

We confirm that RT is a very common event following splenectomy, but in this study it was not associated with clinically evident thrombotic or hemorrhagic complications in children undergoing splenectomy for trauma, structural lesions or non-neoplastic hematological disorders.

## Introduction

Reactive thrombocytosis (RT) refers to thrombocytosis in the absence of a chronic myeloproliferative disease or myelodysplastic disorder. RT is associated with various medical conditions including infections, inflammation, hemorrhage, trauma, or surgery [1]. RT can be either transient or sustained and is driven by increased production of thrombopoietin, catecholamines, and various interleukins (e.g. IL-6 and IL-11) [2].

Postsplenectomy RT is a very common event affecting 75–90% patients undergoing the splenectomy [3, 4]. The platelet counts after splenectomy usually increase 30–100%, with the highest values in the period between 10 to 20 days postoperatively [5]. Thrombocytosis may persist over the next 2–3 months in most cases but in a few patients, thrombocytosis persists indefinitely [6].

Although it is well established that the size and function of platelets in RT are normal and that their interactions with blood vessels remain qualitatively normal [1], RT has been occasionally associated with an increased risk of thrombotic and hemorrhagic events in patients undergoing splenectomy [7, 8]. The thrombogenic potential of RT following splenectomy in children without underlying hematological malignancies (myeloproliferative disorders or myelodysplastic syndromes) are still not clearly defined.

To clarify the incidence and short-term clinical course and possible complications related to RT following splenectomy in the absence of hematological malignancies, we provide a single institution experience with RT in the pediatric population undergoing elective or emergency splenectomy.

## Patients and methods

### Patients and selection criteria

The data of all patients who underwent total splenectomy (urgent or elective) at our pediatric surgical department between January 1999 and December 2018 were retrospectively reviewed. The following data were analyzed: Demographic parameters (age, sex), indications for surgery, operative procedures, preoperative and postoperative platelet count, the use of anticoagulant therapy, and postoperative complications. The patients were followed over the first three weeks postoperatively. According to indications for splenectomy and RT, the patients were divided into the non-neoplastic hematology group and the non-hematology group.

The inclusion criteria for both groups were (1) Patients aged < 16 years; (2) Patients undergoing total splenectomy; (3) Development of postsplenectomy thrombocytosis with platelet count > 500,000/μL; (4) Patients without neoplastic (myeloproliferative and myelodysplastic) hematological disorders.

All the clinical records were fully anonymized prior to the study. The Ethics Committee (Institutional Review Board) of University Clinical Center Sarajevo approved the study and waived informed consent request from the patients and their families (the number of approval: #10-01-2-123/19).

### Definitions and classifications

Total splenectomy was defined as a complete surgical removal of the spleen. Myeloproliferative disorders were defined as any of several hematologic malignancies marked by the excess of one or more types of white blood cells [9]. These disorders include polycythemia vera, essential thrombocytosis, chronic myeloid leukemia (CML), and idiopathic myelofibrosis [9]. Myelodysplastic syndromes were defined as any of a broad group of bone marrow disorders that are characterized by an abnormal reduction in one or more lineages of circulating white blood cells due to defective growth and maturation of blood-forming cells in the bone marrow with a potential to progress to acute myelogenous leukemia (AML).

Thrombocytosis was defined as a platelet count >500,000/μL. Mild thrombocytosis was defined as a platelet count >450,000/μL and <600,000/μL, moderate thrombocytosis a platelet count >600,000/μL and <1,000,000/μL, and extreme thrombocytosis as a platelet count >1,000,000/μL. The platelets were consecutively measured first ten postoperative days after which three additional measurements (11th, 14th and 21st postoperative day) were done.

In an attempt to determine whether an association exists between RT and thrombotic or hemorrhagic complications, one month after hospital discharge all the patients were contacted for further investigations.

All platelet counts were performed on ABBOTT CELL DYN 3700 (Chicago, Illinois, USA) counter, using EDTA anticoagulated fresh blood. The normal range of platelet counts for this machine is 150,000 to 450,000/μL.

### Statistical analysis

To evaluate the normality of the distribution of continuous variables, we used the Shapiro-Wilk (SW) test. The student's t-test was used to compare independent continuous variables that followed the normal distribution. The Wilcoxon signed-rank test was used to test the differences of paired data without the normal distribution. *P*-values < 0.05 were considered significant. All the statistical analyses were performed using the Statistical Package for the Social Sciences (SPSS) IBM Version 26 (SPSS) (UNICOM Systems, Inc.).

### Results

The characteristics of the cohort are summarized in Table 1.

Fifty-two children with a mean age of 9.2 years (range, 0.6–14.5 years) had splenectomy at our institution in the period 1999–2018. All patients were treated with an open splenectomy approach and all received prophylactic antibiotics and pneumococcal vaccine. Among the splenectomy children, 37 (71%) were male and 15 (29%) female patients (male/female ratio of 2.5:1). Severe spleen trauma (rupture) was the most common indication for splenectomy (29 patients, 55.8%), followed by non-neoplastic hematological disorders (18 patients, 34.6%). Three patients (5.8%) had splenectomy for splenic hydatid cysts, and two (3.8%) had splenectomy for multiple splenic abscesses. Among the patients with non-neoplastic hematological

**Table 1. The patients' demographics, their perioperative characteristics, and indications for splenectomy.**

| Variable | Overall | Indications for splenectomy | | P-value |
|---|---|---|---|---|
| | | Non-neoplastic hematology group | Non-hematology group | |
| The number of patients no. (%) | 52 (100%) | 18 (34.6%) | 34 (65.4%) | - |
| Mean age at surgery (yr) | 9.0±1.84 | 9.2±1.22 | 8.6±2.08 | 0.12 |
| Sex (% male) | 37 (71%) | 30 (81%) | 7 (19%) | 0.09 |
| Preoperative platelet count | 324 K/µL | 332.5 K/µL | 317.5 K/µL | 0.98 |
| Short-term adverse events (≤ 3 weeks) (wound infection, wound hematoma and splenic fossa collection) | 4 (7.7%) | 1 (5.6%) | 3 (8.8%) | - |

K/µL, kilo per microliter.

disorders, splenectomy was indicated for hereditary spherocytosis (HS) (n = 16) and immune thrombocytopenia (ITP) (n = 2). Four patients (7.7%) developed postoperative non-thrombotic complications including wound infection, wound hematoma and splenic fossa collection (Table 1). None of the patients developed thrombotic or hemorrhagic complications during the follow-up period. Neither postoperative death nor subsequent fatal infections were observed in the cohort. Forty-nine patients (94.2%) developed a postplenectomy RT. The percentages of mild, moderate and extreme thrombocytosis were 48.9%, 30.7%, and 20.4%, respectively. The comparisons between RT patients in non-neoplastic hematology and non-hematology groups is shown in Table 2. In the non-neoplastic hematology group of patients, a gradual increase in platelet count was observed from the first to sixth day after splenectomy. After this period, the platelet count was gradually decreasing (Table 3). In the non-hematology group of patients, the platelet count decreased in the first two days after splenectomy, probably because of hemodilution but gradually increases starting at day three. From the sixth to the eleventh day, RT has been reported in this group of patients, followed by a gradual decrease in the platelet count, but RT persisted three weeks following splenectomy (Table 3). Although the recorded platelet counts were greater and lasted longer in the non-hematological group compared with the non-neoplastic hematological group, the difference was not significant

**Table 2. The comparative analysis of various clinical parameters between the two groups of the patients affected by reactive thrombocytosis.**

| Variable | Overall | Postplenectomy reactive thrombocytosis | | |
|---|---|---|---|---|
| | | Non-neoplastic hematology group | Non-hematology group | P-value |
| The number of patients no. (%) | 49 (100%) | 16 (32%) | 33 (68%) | - |
| Mean age at surgery (yr) | 9.0±1.9 | 9.3±1.3 | 8.6±2.11 | 0.85 |
| Sex (% male) | 35 (71%) | 29 (82%) | 6 (18%) | 0.08 |
| Preoperative leukocyte count | $12.54 \times 10^9$/L | $9.43 \times 10^9$/L | $13.42 \times 10^9$/L | 0.13 |
| Preoperative platelet count | 329 | 333 | 325 | 0.91 |
| Postoperative leukocyte count (K/µL)* | $12.21 \times 10^9$/L | $11.4 \times 10^9$/L | $12.48 \times 10^9$/L | 0.52 |
| Postoperative platelet count (K/µL)* | 716 | 586 | 846 | 0.47 |
| Mild thrombocytosis no. (%) | 24 (48.9%) | 10 (42%) | 14 (58%) | - |
| Moderate thrombocytosis no. (%) | 15 (30.7%) | 4 (26%) | 11 (74%) | - |
| Extreme thrombocytosis no. (%) | 10 (20.4%) | 2 (20%) | 8 (80%) | - |
| Length of stay (day) | 9.7 | 10.6 | 10.4 | 0.89 |

K/µL, kilo per microliter.

*The presented numbers are mean values that were recorded for these variables in the first 21 postoperative days.

**Table 3. The comparison of platelet counts between the non-neoplastic hematology *vs.* non-hematology group.**

| Variable | Postplenectomy days | | | | | | | | | | | | | |
|---|---|---|---|---|---|---|---|---|---|---|---|---|---|---|
| | 0 | 1 | 2 | 3 | 4 | 5 | 6 | 7 | 8 | 9 | 10 | 11 | 14 | 21 |
| Platelet count in the non-neoplastic hematology group (K/µL) | 294 | 347 | 384 | 448 | 479 | 609 | 675 | 502 | 505 | 511 | 497 | 485 | 460 | 420 |
| Platelet count in the non-hematology group (K/µL) | 343 | 314 | 322 | 378 | 376 | 497 | 619 | 749 | 951 | 974 | 985 | 992 | 760 | 665 |

K/µL, kilo per microliter.

(p = 0.21). Given that the international normalized ratio (INR) values were normal in all patients with extreme RT, the majority of the patients with extreme RT (80%) did not receive any antiplatelet or anticoagulant therapy. Only two patients received the treatment; one child received a low-dose aspirin (5 mg/kg) until their platelet counts fell to the reference values while the older pubertal child received a subcutaneous enoxaparin treatment.

## Discussion

It is well known that the incidence of thrombocytosis shows an age-dependent pattern with the highest reported occurrence in infants aged up to 24 months [10, 11]. Specifically, during childhood, platelet counts greater than 500,000/µL may be seen in 13% of neonates at birth, 36% during the first month, particularly in low birth-weight infants, and 13% in 6 to 11 months of age [11]. The trend of a gradual decrease in platelet counts continues throughout childhood and reaches normal levels by the age of 11 [11]. Thrombocytosis in children is caused by the immaturity of their innate and/or adaptive immunity or more frequently by infections [12]. We found no statistically significant difference in the age of children with RT because splenectomized children in our cohort were on average older (9.2 years) and because of the cause of thrombocytosis itself.

RT in childhood is usually associated with bacterial or viral infections; less frequent causes include anemias (hemolytic or iron-deficiency), autoimmune diseases, cancers and drugs [11]. Since the spleen in healthy persons is a reservoir for approximately 1/3 of total body platelets, splenectomy is a well-recognized cause of RT [3, 5, 13, 14]. Our study revealed a very high (94%) prevalence of RT among the pediatric population undergoing splenectomy. Although this prevalence appears to be high, it is in line with the previous studies that have been reported in pediatric population [1, 3, 4, 13, 14]. In addition, splenectomy was found to be one of the main causes of extreme RT [15]. In our study, the extreme thrombocytosis accounted for 20% of all reported cases of postsplenectomy RT.

Another important finding from our study is a higher incidence of RT including extreme RT in male patients, with the male/female ratio 2.5:1, which is in agreement with the previous studies [16–18]. The reasons for male predominance can be partially explained by the synergistic effects of sex hormones, especially androgens, on thrombopoiesis and androgen modulation of platelet count and function [19].

Extreme thrombocytosis may predispose several complications including neurological (vasomotor), thrombotic, and hemorrhagic [20]. However, in case of postsplenectomy RT, the platelets retain their normal structure and function, the bone marrow is normal, and the interaction of platelets with the vessel wall remains normal. Therefore, the association of thrombocytosis with complications such as thrombosis and hemorrhage appear to be related to quantitative rather than qualitative platelet abnormalities [1, 20]. This was also confirmed in our study given that the measured INR was normal in all cases including extreme RT. However, thrombotic complications have been occasionally reported in pediatric RT following splenectomy for other reasons [21, 22].

Some observations suggest that leukocytosis may also increase the risk of thrombosis in patients with myeloproliferative disorders [23], but no such association was found in patients without myeloproliferative disorders. Among our patients, there were no cases of clinically suggestive thrombosis including cases of portal splenic vein thrombosis (PSVT).

Previous studies have suggested that fatal thrombotic complications in adult patients with postplenectomy thrombocytosis can be prevented by prophylactic anticoagulants or antiplate-let drugs [24]. In contrast, studies that are more recent have shown a much lower rate of thrombotic events in postsplenectomy adult patients, which did not differ significantly compared to other postoperative groups [25]. The exceptions are patients with myeloproliferative neoplasms in whom platelets are often morphologically and functionally abnormal with the consequent increased risk of thrombosis and bleeding [26].

The treatment of asymptomatic infants and children with extreme RT is still controversial. However, 80% of children with extreme RT in our study did not receive any antiplatelet or anticoagulant therapy and had no any adverse event. These findings further support previous studies that revealed a low risk of complications among this population [1, 27–29].

Our study has several limitations including its retrospective nature, single center experience and small sample size. A longer follow-up period would provide valuable information on potentially late thrombotic events among the postplenectomy children with persistent RT. In addition, a postoperative ultrasound scans were not part of the routine checkups for occult splenic and portal vein thrombosis, which may affect these patients [22]. Despite all the limitations, we confirm that RT is a very common event following splenectomy, but in this study it was not associated with clinically evident thrombotic or hemorrhagic complications in children undergoing splenectomy for trauma, structural lesions or non-neoplastic hematological disorders.

## Author Contributions

**Conceptualization:** Zlatan Zvizdic, Semir Vranic.

**Data curation:** Zlatan Zvizdic, Aladin Kovacevic, Emir Milisic, Asmir Jonuzi, Semir Vranic.

**Formal analysis:** Zlatan Zvizdic, Semir Vranic.

**Funding acquisition:** Semir Vranic.

**Methodology:** Zlatan Zvizdic, Semir Vranic.

**Resources:** Semir Vranic.

**Supervision:** Semir Vranic.

**Validation:** Zlatan Zvizdic, Aladin Kovacevic, Emir Milisic, Asmir Jonuzi, Semir Vranic.

**Writing – original draft:** Zlatan Zvizdic, Semir Vranic.

**Writing – review & editing:** Zlatan Zvizdic, Semir Vranic.

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
