## [Decision Letter · Decision Letter 0]

8 Jun 2020

PONE-D-20-07037

Clinical course and short-term outcome of postsplenectomy reactive thrombocytosis in children without myeloproliferative disorders: A single institutional experience from a developing country

PLOS ONE

Dear Dr. Vranic,

Thank you for submitting your manuscript to PLOS ONE. After careful consideration, we feel that it has merit but does not fully meet PLOS ONE’s publication criteria as it currently stands. Therefore, we invite you to submit a revised version of the manuscript that addresses the points raised during the review process.

The article is an interesting issue. However, it cannot be accepted at its current content because it cannot fulfill the following acceptance criteria " Conclusions are presented in an appropriate fashion and are supported by the data." and "The article is presented in an intelligible fashion and is written in standard English. " 

Our reviewers raised several important comments. We look for the author could specifically reply it and make corresponding revision appropriately in response to the comments. 

We look forward to receiving your revised manuscript.

Kind regards,

Chun Chieh Yeh, M.D., Ph.D.

Academic Editor

PLOS ONE

Journal Requirements:

2. In the ethics statement in the manuscript and in the online submission form, please provide additional information about the patient records used in your retrospective study. Specifically, please ensure that you have discussed whether all data were fully anonymized before you accessed them and/or whether the IRB or ethics committee waived the requirement for informed consent. If patients' guardians provided informed written consent to have data from their medical records used in research, please include this information.

Additional Editor Comments (if provided):

The article is an interesting issue. However, it cannot be accepted at its current content because it cannot fulfill the following acceptance criteria " Conclusions are presented in an appropriate fashion and are supported by the data." and "5. The article is presented in an intelligible fashion and is written in standard English. "

Our reviewer raised several important comments. We look for the author could specifically reply it and make corresponding revision appropriately in response to the comments.

Reviewers' comments:

Reviewer's Responses to Questions

**Comments to the Author**

1. Is the manuscript technically sound, and do the data support the conclusions?

Reviewer #1: Yes

Reviewer #2: Yes

2. Has the statistical analysis been performed appropriately and rigorously? 

Reviewer #1: Yes

Reviewer #2: Yes

3. Have the authors made all data underlying the findings in their manuscript fully available?

Reviewer #1: No

Reviewer #2: Yes

4. Is the manuscript presented in an intelligible fashion and written in standard English?

Reviewer #1: Yes

Reviewer #2: No

5. Review Comments to the Author

Reviewer #1: This is a retrospective single institution review of 52 pediatric open splenectomies, divided into non-hematologic (n=34) and hematologic non-neoplastic (n=18) indications. Ninety four percent of the sample developed post-splenectomy thrombocytosis but no thrombotic complications were reported.

The article is generally well written. However, important data are missing and the conclusion that post-splenectomy reactive thrombocytopenia “…is not associated with thrombotic or hemorrhagic complications in children without underlying haematological malignancies” is inadequately supported by the data (see below).

1. Severe spleen trauma was commonest indication for splenectomy (56%) – this is surprising as >90% of blunt pediatric splenic trauma can be successfully managed conservatively. Perhaps this was penetrating trauma?

2. What was the protocol for recording postoperative platelet counts? Table 2 values are not meaningful unless they refer to peak postoperative platelet counts and the accuracy of the data depends on the frequency of blood sampling. Presumably, the figures in this table are means?

3. There are no data on spleen size, which is a factor in postoperative thrombotic complications.

4. Did any of the patients receive any postoperative anticoagulant or antiplatelet therapy? The discussion suggests that 20% of the extreme thrombocytosis group did receive treatment.

5. The authors only report clinically evident thrombotic complications. No routine postoperative ultrasound scans were performed to check for occult splenic and portal vein thrombosis (see Stringer MD, Lucas N. Thrombocytosis and portal vein thrombosis after splenectomy for paediatric haemolytic disorders: How should they be managed? J Paediatr Child Health. 2018;54(11):1184-1188).

6. Figure 1 axis labelling is incomplete

Reviewer #2: Title:

Clinical course and short-term outcome of postsplenectomy reactive thrombocytosis in children without myeloproliferative disorders: A single institutional experience from a developing country

Outlines:

The authors conduct a retrospective review of 52 children aged younger than 16 years with postsplenectomy reactive thrombocytosis. They found postsplenectomy reactive thrombocytosis is not associated thrombotic or hemorrhagic complications in children without underlying hematological malignancies.

The following are some comments and questions for the authors:

1. Page 8, line 170. The authors mentioned the platelet count decreased in patients in the non-neoplastic hematology group. But in Table 3, it was reversed. The platelet count increased in patients in the non-neoplastic hematology group after postsplenectomy day 1. Please clarify for the readers.

2. Page 8, line 172-173. The authors mentioned that RT has reported in the non-neoplastic hematology group from 5th to 10th day. But in Table 3, thrombocytosis (>500,000/uL) was observed from postsplenectomy day 3 to 9. Please clarify for the readers.

3. Page 9, line 175-176. The authors mentioned that maximal platelet count was recorded on the 14th day. But in Table 3, the highest platelet count in the non-hematology group is on postsplenectomy day 11. Please clarify for the readers.

4. I can’t understand Figure 1. What does X-axis indicate? What does Y-axis indicate? How to interpret the relationship between INR and platelet count based on the present figure (only 2 points and 1 line)?

5. Page 11, line 226. Please cite references about postsplenectomy thromboembolic complications in children.

6. There are several language flaws in the article needs to be polished further.

6. PLOS authors have the option to publish the peer review history of their article (what does this mean?). If published, this will include your full peer review and any attached files.

Reviewer #1: No

Reviewer #2: No

---

## [Author Response · Author response to Decision Letter 0]

13 Jun 2020

Reviewer #1: 

This is a retrospective single institution review of 52 pediatric open splenectomies, divided into non-hematologic (n=34) and hematologic non-neoplastic (n=18) indications. Ninety four percent of the sample developed post-splenectomy thrombocytosis but no thrombotic complications were reported.

The article is generally well written. However, important data are missing and the conclusion that post-splenectomy reactive thrombocytopenia “…is not associated with thrombotic or hemorrhagic complications in children without underlying haematological malignancies” is inadequately supported by the data (see below).

1. Severe spleen trauma was commonest indication for splenectomy (56%) – this is surprising as >90% of blunt pediatric splenic trauma can be successfully managed conservatively. Perhaps this was penetrating trauma?

ANSWER: Thanks for your comment. We agree with your statement that blunt pediatric splenic trauma can be successfully treated conservatively in most cases. Such conservative or non-operative approach has been applied at our institution for many years with excellent results. Penetrating trauma caused by shrapnel from unexploded ordnance from the war of the 1990s in our country was the cause of splenectomy in four children. These children had multiple injuries to the organ systems. In one child, a spleen injury was caused by a knife blade. However, the remaining 24 children who underwent splenectomy over a twenty-year period had serious splenic injuries that resulted in unsuccessful conservative treatment.

2. What was the protocol for recording postoperative platelet counts? Table 2 values are not meaningful unless they refer to peak postoperative platelet counts and the accuracy of the data depends on the frequency of blood sampling. Presumably, the figures in this table are means?

ANSWER: You are correct; the presented numbers are mean values. We now added an “*” with the legend that refers to these numbers. Regarding the protocol for postoperative platelet counts, these were recorded first ten postoperative days in a consecutive manner, after which three additional measurements were done (11th, 14th and 21st day). We now updated this in Materials and Methods (page 7, lines 137-138).

3. There are no data on spleen size, which is a factor in postoperative thrombotic complications.

ANSWER: Thanks for bringing this important issue to our attention. We have not provided the spleen size info in the manuscript given that we have incomplete spleen size measurements and all were done preoperatively. 

4. Did any of the patients receive any postoperative anticoagulant or antiplatelet therapy? The discussion suggests that 20% of the extreme thrombocytosis group did receive treatment.

ANSWER: Two patients received the treatment; one child received a low‐dose aspirin (5 mg/kg) until their platelet counts fell to reference values while the older pubertal child received a subcutaneous enoxaparin. The results paragraph have been updated accordingly (page 9, lines 190-192).

5. The authors only report clinically evident thrombotic complications. No routine postoperative ultrasound scans were performed to check for occult splenic and portal vein thrombosis (see Stringer MD, Lucas N. Thrombocytosis and portal vein thrombosis after splenectomy for paediatric haemolytic disorders: How should they be managed? J Paediatr Child Health. 2018;54(11):1184-1188).

ANSWER: Thanks for your suggestion. Although asymptomatic cases of acute postoperative splenic and portal vein thrombosis detected by Doppler ultrasound have been well documented in children, postoperative Doppler ultrasound scanning in children undergoing splenectomy due to hemolytic diseases has recently been introduced at our institution. For this reason, we were able to analyze the possible existence of only clinically evident cases of thrombotic complications. We addressed your concerns in our limitations of the study (page 12, lines 252-254).

6. Figure 1 axis labelling is incomplete

ANSWER: We decided to remove the Figure 1 given your and the comment from another reviewer.

Reviewer #2: Title:

Clinical course and short-term outcome of postsplenectomy reactive thrombocytosis in children without myeloproliferative disorders: A single institutional experience from a developing country

Outlines:

The authors conduct a retrospective review of 52 children aged younger than 16 years with postsplenectomy reactive thrombocytosis. They found postsplenectomy reactive thrombocytosis is not associated thrombotic or hemorrhagic complications in children without underlying hematological malignancies.

The following are some comments and questions for the authors:

1. Page 8, line 170. The authors mentioned the platelet count decreased in patients in the non-neoplastic hematology group. But in Table 3, it was reversed. The platelet count increased in patients in the non-neoplastic hematology group after postsplenectomy day 1. Please clarify for the readers.

ANSWER: We have corrected it.

2. Page 8, line 172-173. The authors mentioned that RT has reported in the non-neoplastic hematology group from 5th to 10th day. But in Table 3, thrombocytosis (>500,000/uL) was observed from postsplenectomy day 3 to 9. Please clarify for the readers.

ANSWER: We have now corrected in the text. 

3. Page 9, line 175-176. The authors mentioned that maximal platelet count was recorded on the 14th day. But in Table 3, the highest platelet count in the non-hematology group is on postsplenectomy day 11. Please clarify for the readers.

ANSWER: We have corrected it in the text.

4. I can’t understand Figure 1. What does X-axis indicate? What does Y-axis indicate? How to interpret the relationship between INR and platelet count based on the present figure (only 2 points and 1 line)?

ANSWER: We decided to remove the figure 1 given your and the comment from another reviewer.

5. Page 11, line 226. Please cite references about postsplenectomy thromboembolic complications in children.

ANSWER: The references have been added (refs#21 and 22).

6. There are several language flaws in the article needs to be polished further.

ANSWER: Thank you for your comment. We have done additional proofreading to make the manuscript fully compatible with academic English standards.

---

## [Decision Letter · Decision Letter 1]

9 Jul 2020

PONE-D-20-07037R1

Clinical course and short-term outcome of postsplenectomy reactive thrombocytosis in children without myeloproliferative disorders: A single institutional experience from a developing country

PLOS ONE

Dear Dr. Vranic,

Thank you for submitting your manuscript to PLOS ONE. After careful consideration, we feel that it has merit but does not fully meet PLOS ONE’s publication criteria as it currently stands. Therefore, we invite you to submit a revised version of the manuscript that addresses the points raised during the review process.

I wish you could make appropriate response to the reviewer-1 comments. Decision will be made after your revision.

We look forward to receiving your revised manuscript.

Kind regards,

Chun Chieh Yeh, M.D., Ph.D.

Academic Editor

PLOS ONE

Additional Editor Comments (if provided):

Thanks to your appropriate response and revision to the reviewers' comments. The reviewer-1 still raised a suggestion for minor revision. I wish you could consider and make appropriate response to this suggestion. Decision will be made based on your response. Thanks.

Reviewers' comments:

Reviewer's Responses to Questions

**Comments to the Author**

1. If the authors have adequately addressed your comments raised in a previous round of review and you feel that this manuscript is now acceptable for publication, you may indicate that here to bypass the “Comments to the Author” section, enter your conflict of interest statement in the “Confidential to Editor” section, and submit your "Accept" recommendation.

Reviewer #1: All comments have been addressed

Reviewer #2: All comments have been addressed

2. Is the manuscript technically sound, and do the data support the conclusions?

Reviewer #1: No

Reviewer #2: Yes

3. Has the statistical analysis been performed appropriately and rigorously? 

Reviewer #1: Yes

Reviewer #2: Yes

4. Have the authors made all data underlying the findings in their manuscript fully available?

Reviewer #1: Yes

Reviewer #2: Yes

5. Is the manuscript presented in an intelligible fashion and written in standard English?

Reviewer #1: Yes

Reviewer #2: Yes

6. Review Comments to the Author

Reviewer #1: The authors have responded appropriately to my comments BUT their conclusions are based on just 52 patients and are invalid. They did NOT routinely use postoperative ultrasound assessment of the portal and splenic vein nor did they record data on spleen size.

Nevertheless, I think the article would be valid if the conclusions are changed in both the Abstract and Discussion.

The following is suggested:

“We confirm that RT is a very common event following splenectomy, but IN THIS STUDY IT WAS not associated with CLINICALLY EVIDENT thrombotic or hemorrhagic complications in children undergoing splenectomy for trauma, structural lesions or non-neoplastic hematological disorders.”

Reviewer #2: Thanks the authors for the revision. The manuscript looks better and more clear to me. I have no other comments.

7. PLOS authors have the option to publish the peer review history of their article (what does this mean?). If published, this will include your full peer review and any attached files.

Reviewer #1: No

Reviewer #2: No

---

## [Author Response · Author response to Decision Letter 1]

9 Jul 2020

COMMENTS:

Reviewer #1: The authors have responded appropriately to my comments BUT their conclusions are based on just 52 patients and are invalid. They did NOT routinely use postoperative ultrasound assessment of the portal and splenic vein nor did they record data on spleen size.

Nevertheless, I think the article would be valid if the conclusions are changed in both the Abstract and Discussion.

The following is suggested:

“We confirm that RT is a very common event following splenectomy, but IN THIS STUDY IT WAS not associated with CLINICALLY EVIDENT thrombotic or hemorrhagic complications in children undergoing splenectomy for trauma, structural lesions or non-neoplastic hematological disorders.”

Answer: Thank you for your favorable comments.

We also agree with your observation and the conclusion in both the abstract and discussion paragraph has been revised accordingly (lines 51-4 and 248-51).

Reviewer #2: Thanks the authors for the revision. The manuscript looks better and more clear to me. I have no other comments.

Answer: Thank you.

---

## [Editor Report · Decision Letter 2]

20 Jul 2020

Clinical course and short-term outcome of postsplenectomy reactive thrombocytosis in children without myeloproliferative disorders: A single institutional experience from a developing country

PONE-D-20-07037R2

Dear Dr. Vranic,

We’re pleased to inform you that your manuscript has been judged scientifically suitable for publication and will be formally accepted for publication once it meets all outstanding technical requirements.

Kind regards,

Chun Chieh Yeh, M.D., Ph.D.

Academic Editor

PLOS ONE

Additional Editor Comments (optional):

Thanks for your appropriate response specific to reviewers' comments. I think the work is worthy of acceptance. Congratulation.
---

## [Editor Report · Acceptance letter]

21 Jul 2020

PONE-D-20-07037R2 

Clinical course and short-term outcome of postsplenectomy reactive thrombocytosis in children without myeloproliferative disorders: A single institutional experience from a developing country 

Dear Dr. Vranic:

I'm pleased to inform you that your manuscript has been deemed suitable for publication in PLOS ONE. Congratulations! Your manuscript is now with our production department. 

Kind regards, 

on behalf of

Dr. Chun Chieh Yeh 

Academic Editor

PLOS ONE